# Index Coding with Multiple Interpretations

**DOI:** 10.3390/e24081149

**Published:** 2022-08-18

**Authors:** Valéria G. Pedrosa, Max H. M. Costa

**Affiliations:** School of Electrical and Computer Engineering, University of Campinas, Campinas 13083-852, SP, Brazil

**Keywords:** index coding, pliable index coding, error correcting index coding

## Abstract

The index coding problem consists of a system with a server and multiple receivers with different side information and demand sets, connected by a noiseless broadcast channel. The server knows the side information available to the receivers. The objective is to design an encoding scheme that enables all receivers to decode their demanded messages with a minimum number of transmissions, referred to as an index code length. The problem of finding the minimum length index code that enables all receivers to correct a specific number of errors has also been studied. This work establishes a connection between index coding and error-correcting codes with multiple interpretations from the tree construction of nested cyclic codes. The notion of multiple interpretations using nested codes is as follows: different data packets are independently encoded, and then combined by addition and transmitted as a single codeword, minimizing the number of channel uses and offering error protection. The resulting packet can be decoded and interpreted in different ways, increasing the error correction capability, depending on the amount of side information available at each receiver. Motivating applications are network downlink transmissions, information retrieval from datacenters, cache management, and sensor networks.

## 1. Introduction

In this work, we consider a source code variant, introduced by Birk and Kol [1], originally called informed source coding-on-demand (ISCOD), and further developed by Bar-Yossef et al. [2]. Motivating applications include satellite transmission of large files, audio and video on demand (such as streaming networks), database data retrieval, cache management for network applications and sensor networks. The model considered in [1] involves a source that possesses *n* messages and *m* receivers. Each receiver knows a proper subset of messages, which is referred to as the side information and demands a specific message unknown to it. The source, aware of the messages possessed by each receiver, uses this knowledge to develop a transmission scheme that satisfies the demands of all receivers using as few transmissions as possible, referred to as the index code length.

Index coding can be viewed as special case of rate distortion with multiple receivers, each with some side information about the source [3]. Index coding has received considerable attention recently, motivated by applications in multi-user broadcast scenarios, such as audio and video on demand, streaming networks, satellite communications and by its connection to network coding. In [4,5], the equivalence between network encoding and index encoding has been established. This research topic has been extended in other directions, such as pliable index coding [6], a variation of index coding in which we still consider a server and *m* clients with side information, but where the receivers are *flexible* and satisfied to receive any message that is not in their side information set; such flexibility can reduce the amount of communication, sometimes significantly. This has applications in music streaming services or internet searching, such as *content distribution networks* (CDNs) [7]; a CDN manages servers in multiple geographically distributed locations, stores copies of the web content (including documents, images, audio and others) in its servers and attempts to direct each user request to a CDN location that will provide the best user experience. In this application, each receiver may be interested in receiving any message that it does not already possess as side information. Suppose that we are searching for the latest news and we already have some information. We are happy if we obtain any additional news that we do not have, with minimum delay. Here, we do not specify the news. On a music streaming service, users do not know which song will play next; they are usually only guaranteed that it will be one of a certain group and that it will not be repeated. In online advertising systems, customers do not require a specific advertisement to view; it is the distributor who chooses which one will be placed on customers’ screens. The distributor may wish to avoid repeating the same advertisement for the same customer, as this can decrease customer satisfaction.

How much we can gain in terms of bandwidth and user satisfaction, if recommendation systems become bandwidth-aware and take into account not only the user preferences? Song and Fragouli [8] formulated this as a new problem in the context of index coding, where they relaxed the index coding requirements and considered the case where the customer is satisfied to receive any message that they do not already have, with satisfaction proportional to their preference for that message.

A promising research area that has recently emerged is in how to use index coding to improve the communication efficiency in distributed computing systems, especially for data shuffling in iterative computations [9,10]. Index coding has been proposed to increase the efficiency of data shuffling, which can form a major communication bottleneck for big data applications. In particular, pliable index coding can offer a more efficient framework for data shuffling, as it can better leverage the many possible shuffling choices to reduce the number of transmissions.

The index coding problem subject to transmission errors was initially considered by Dau et al. [11]. In this work, we establish a connection between index coding and error-correcting codes with multiple interpretations from the tree construction of nested cyclic codes proposed in [12]. The notion of multiple interpretation using nested codes [13] is as follows: multiple information packets are separately encoded via linear channel codes, and then combined by addition and transmitted as a single codeword, minimizing the number of channel uses and offering error protection. The resulting packet can be decoded and interpreted in different ways, yielding an increase in error correction capability, depending on the amount of side information available at each receiver.

Part of the content of this paper was presented in [14]. In the current version, evidence to verify our claims has been added, as well as some examples. The results in this paper are an extension of the results in [12,14].

The main contributions of this paper are as follows.

We verify that, for cyclic codes, there will not always be an increase in error correction capability between different levels of the code tree. For this reason, we initially restrict the study to Reed–Solomon codes since they are maximum separable distance (MDS) codes, and provide an increase in Hamming distance at each level. This means that, under certain conditions, knowledge of side information can be interpreted as an increase in error correction capability.We propose a new variant for the index coding problem, which we call “index coding with multiple interpretations”. We assume that receivers demand all the messages from the source and that the sender is unaware of the subset of messages already known by the receivers. The sender performs encoding such that any side information may be used by the decoder in order to increase its error correction capability. Moreover, if a receiver has no side information, the decoder considers the received word to belong to the highest rate code, associated with the root node of the tree.We also propose a solution to relax some constraints on how side information should occur at the receivers, using graph coloring associated with the pliable index coding problem.

## 2. Preliminaries

### 2.1. Notation and Definitions

For any positive integer *n*, we let [n]:={1,…,n}. We write $F_{q}$ to denote the finite field of size *q*, where *q* is a prime power, and use ${F_{q}}^{n\times t}$ to denote the vector space of all n×t matrices over $F_{q}$.

### 2.2. Review of Linear and Cyclic Codes

We now introduce the notation and briefly review some of the relevant properties of linear and cyclic codes based on [15,16]. The purpose of a code is to add extra check symbols to the data symbols so that errors may be found and corrected at the receiver. That is, a sequence of data symbols is represented by some longer sequence of symbols with enough redundancy to protect the data. In general, to design coding schemes for receivers with side information, we will consider collections of linear codes that are of length *n* over $F_{q}$.

#### Structure of Linear Block Codes

Recall that under componentwise vector addition and componentwise scalar multiplication, the set of *n*-tuples of elements from $F_{q}$ is the vector space called $F_{q}^{n}$. For the vectors $u=\left(u_{1},\ldots ,u_{n}\right)\in F_{q}^{n}$ and $v=\left(v_{1},\ldots ,v_{n}\right)\in F_{q}^{n}$, the *Hamming* distance between *u* and *v* is defined to be the number of coordinates *u* and *v* that differ, i.e.,
$$d\left(u,v\right)=|\{i\in \left[n\right]:u_{i}\neq v_{i}\}|.$$

**Definition** **1.**
*A k-dimensional subspace C of $F_{q}^{n}$ is called a linear ${\left(n,k,d\right)}_{q}$ code over $F_{q}$ if the minimum distance of C,*

$$d\left(C\right)≜\min_{u,v\in C,u\neq v}d\left(u,v\right)$$

*is equal to d. Sometimes, we only use ${\left(n,k\right)}_{q}$ to refer to the code C, where n is the length of the codewords and k is the dimension of the code. The code’s rate is the ratio $\frac{k}{n}$.*


That is, a ${\left(n,k\right)}_{q}$ linear code C can be completely described by any set of *k* linearly independent codewords $v_{1},v_{2},\ldots ,v_{k}$; thus, any codeword is one of the $q^{k}$ linear combinations $\sum_{i_{1}}^{k}\alpha_{i}v_{i}$, $\alpha_{i}\in F_{q}$. If we arrange the codewords into a k×n matrix *G*, we say that *G* is a *generator matrix* for C.

A special case of major importance is $F_{2}^{n}$, which is the vector space of all binary codewords of length *n* with two such vectors added by modulo-2 addition in each component. A binary code of size $M=2^{k}$ for an integer *k* is referred to as an (n,k) binary code.

We consider cyclic codes of length *n* over $F_{q}$ with gcd(n,q)=1. Label the coordinates of $c\in F_{q}^{n}$ with the elements of $Z_{n}=\left\{0,1\ldots ,n-1\right\}$ and associate the vector $c=(c_{0},\ldots ,c_{n-1})$ with the polynomial $c\left(x\right)=c_{0}+c_{1}x+\cdots +c_{n-1}x^{n-1}$. With this correspondence, a cyclic code C is an ideal of the ring $R_{n}=F_{q}\left[x\right]/\left(x^{n}-1\right)$. We use g(x) to denote the generator polynomial of C and write $C=\langle g\left(x\right)\rangle =\left\{C\left(x\right)\in F_{q}\left[x\right];g\left(x\right)|C\left(x\right)\right\}$ to describe a *t*-error correcting cyclic code.

### 2.3. Index Coding with Side Information

The system shown in Figure 1 illustrates the index coding problem. Receiver $R_{i}$ is requesting the message $x_{i}$, i∈{1,2,3} and knows other messages as side information; $R_{1}$ knows $x_{3}$, $R_{2}$ knows $x_{1}$ and $x_{3}$ and the receiver $R_{3}$ knows $x_{1}$ and $x_{2}$.

The goal of index coding is to perform the joint encoding of the messages, in order to simultaneously meet the demands of all receivers, while transmitting the resulting messages at the highest possible rate.

Assuming a noiseless broadcast channel, the server would communicate all messages by sending one at a time, in three transmissions.

Alternatively, when transmitting the two coded messages $x_{1}$ and $x_{2}⊕x_{3}$, the receiver $R_{1}$ decodes $x_{1}$, from $\left(x_{2}⊕x_{3}\right)⊕x_{3}=x_{2}$ and $\left(x_{2}⊕x_{3}\right)⊕x_{2}=x_{3}$, $R_{2}$ and $R_{3}$ recover their demands.

The index coding problem is formulated as follows. Suppose that a server *S* wants to send a vector $x=(x_{1},x_{2},\ldots ,x_{n})$, where $x_{i}\in F_{q}\forall i\in \left[n\right]$, to [n] receivers $R_{1},R_{2},\ldots ,R_{n}$. Each receiver $R_{i}$ has $x_{S_{i}}=\left\{x_{j};j\in S_{i}\subseteq \left[n\right]\setminus \left\{i\right\}\right\}$ as side information and is interested in receiving the message $x_{i}$. The codeword $C\left(x\right)\in F_{q}^{\ell }$ is sent and allows each receiver $R_{i}$ to retrieve $x_{i}$. C is an index code *scalar* over $F_{q}$ of length *ℓ*. The purpose of *S* is to find an index code that has the minimum length. The index code is called linear if C(x) is a linear function.

#### Index Coding via Fitting Matrices

A directed graph G=(V,E) with *n* vertices specifies an instance of the index coding problem. Each vertex of G corresponds to a receiver (and its demand) and there is a directed edge i⟶j if and only if the receiver $R_{i}$ knows $x_{j}$ as side information. Then, we write:$$S_{i}=\left\{j:\left(i,j\right)\;\mathrm{is}\;a\;\mathrm{edge}\;\mathrm{of}\;G\right\}$$

**Definition** **2.**
*Let G=(V,E) be a directed graph on n vertices without self-loops.*


*1*.
*A 0-1 matrix $M=(m_{ij})$, whose rows and columns are labeled by the elements of V=[n], fits G if, for all i and j,*
(i)
*$m_{ii}=1$;*
(ii)
*For i≠j,*

$$m_{ij}=\left\{\begin{array}{cc} {}*\in \{0,1\}; & if(i,j)\;is\;an\;edge\;of\;G;\\ 0 & ;else. \end{array}\right.$$


*Thus, M−I is the adjacency matrix of an edge-induced subgraph of G, where I denotes the n×n identity matrix.*
*2*.
*The minrank of G over the field $F_{2}$ is defined as follows:*

$${minrk}_{2}\left(G\right)≜min\left\{{rank}_{2}\left(M\right):MfitsG\right\}$$



**Remark** **1.**
*The term rank${}_{2}\left(M\right)$ denotes the rank of such matrix M over $F_{2}$, after “∗” has been assigned a value of 0 or 1. As an example for the index coding problem instance described in Figure 1, the matrix M would be given as follows:*

$$M=\left[\begin{array}{ccc} 1 & 0 & *\\ {}* & 1 & *\\ {}* & * & 1 \end{array}\right]$$



**Example** **1.**
*Consider the side information graph G and a matrix M that fits G, as shown in Figure 2. As ${minrank}_{2}\left(M\right)=2$, we can select two linearly independent rows in a matrix M, namely $M^{\prime }$, and design an linear index code with the shortest possible length. The codeword sent will be $M^{\prime }x$.*


**Theorem** **1**([2]). *For any side information graph G, there exists a linear index code for G whose length equals minrk${}_{2}\left(G\right)$. This bound is optimal for all linear index codes G.*

In [17], the index encoding problem was generalized. Suppose that a sender wants to transmit *n* messages $\left(X_{1},\ldots ,X_{n}\right)$, where $X_{i}\in {F_{q}}^{t}\forall i\in \left[n\right]$, to *m* receivers $R_{1},\ldots ,R_{m}$, through a noiseless channel. The receiver $R_{j}$ is interested in recovering a single block $X_{f(j)}$, where f:[m]⟶[n], and knows $X_{S_{j}}=\left\{X_{i};i\in S_{j}\subseteq \left[n\right]\setminus f\left(j\right)\right\}$. The goal is to satisfy the demands of all receivers, exploiting their side information in a minimum number of transmissions.

When m=n, f(j)=j,∀j∈[m] and t=1, we have a *scalar index code* [2]. Otherwise, we have a *vector index code*.

Let $I=\{S_{j};j\in \left[m\right]\}$ be the set of side information of all receivers. An instance of an index coding problem given by (m,n,I,f) can be described by a directed hypergraph.

**Definition** **3.**
*The side information hypergraph $H\left(V,E_{H}\right)=H\left(m,n,I,f\right)$ is defined by the set of vertices V=[n] and the (directed) hyperedges $E_{H}$, where*

$$E_{H}=\left\{e_{j}=\left(f\left(j\right),S_{j}\right);j\in \left[m\right]\right\}$$

*A hyperedge $e_{j}=\left(f\left(j\right),S_{j}\right)$ represents the demand and side information of the receiver $R_{j}$.*


**Example** **2.**
*Consider an instance of an index coding problem in Figure 3. The hypergraph in Figure 3b describes the problem, where n=4 (messages) and m=5 (receivers) requiring f(1)=1, f(2)=3, f(3)=4, f(4)=4 and f(5)=2, and with the following side information sets $S_{1}=\left\{3,4\right\},S_{2}=\left\{2,4\right\},S_{3}=\left\{1\right\}$, $S_{4}=\left\{2\right\}$ and $S_{5}=\left\{1,3\right\}$, respectively.*


**Definition** **4.**
*Given an instance of an index encoding problem described by H(m,n,I,f),*

$$C:{F_{q}}^{n\times t}⟶{F_{q}}^{\ell \times t},$$

*is a $F_{q}$- index code with length ℓ, for the instance described by F, if, for each receiver $R_{j}$, j∈[m], there exists a decoding function*

$$D_{j}:{F_{q}}^{\ell \times t}\times {F_{q}}^{t|S_{j}|}⟶{F_{q}}^{t},$$

*satisfying $D_{j}\left(C\left(X\right),X_{S_{j}}\right)=X_{f(j)},\forall X\in {F_{q}}^{n\times t}$.*


The *transmission rate* of the code is defined as $\frac{\ell }{t}$. If t=1, then the index code is known as a *scalar index code*; otherwise, it is known as a *vector index code*. A linear coding function C is also called a linear index code. The goal of index coding is to find optimal index codes, i.e., those with the minimum possible transmission rate. For scalar linear index codes, we refer to the quantity r as the length of the code, and thus rate optimality translates to minimal length codes.

**Definition** **5.**
*C is a $F_{q}$-linear index code, $C\left(X\right)=GX,\forall X\in F_{q}^{\ell \times n}$, where $G\in F_{q}^{\ell \times n}$. G is the matrix that generates the linear index code C.*


The following definition generalizes the minrank definition over $F_{q}$ of the side information graph G, which was defined in [2], to a hypergraph H(m,n,I,f).

**Definition** **6.**
*Let Supp$\left(v\right)=\{i\in \left[n\right]:v_{i}\neq 0\}$, the support of a vector $v\in F_{q}^{n}$. The Hamming weight of v will be denoted by ω(v)=|Supp(v)|, the number of nonzero coordinates of v.*


**Definition** **7**([11]). *Suppose that H(m,n,I,f) corresponds to an instance of index coding with side information (ICSI). Then, the minrank of H over $F_{q}$ is defined as*
$${minrk}_{q}\left(H\right)≜min\left\{{rank}_{q}\left({\left\{v_{i}+e_{f(i)}\right\}}_{i\in [m]}\right):v_{i}\in F_{q}Supp\left(v_{i}\right)\subseteq S_{i}\right\}$$

This may be rewritten as follows.

**Definition** **8.**
*Let a side information hypergraph H correspond to an instance of the ICSI problem. A matrix $M=\left(m_{ij}\right)\in F_{q}^{m\times n}$ fits H if*

$$m_{ij}=\left\{\begin{array}{cc} 1 & if\;j=f(i)\\ 0 & if\;j\in /S_{i} \end{array}\right.$$

*The minrank of F over the field $F_{q}$ is defined as follows:*

$${minrk}_{q}\left(H\right)≜min\left\{{rank}_{q}\left(M\right):MfitsH\right\}$$



**Theorem** **2**([2]). *Given an instance of an index encoding problem described by the hypergraph H(m,n,I,f), the optimal length of an index code on the field $F_{q}$ is ${minrk}_{q}\left(H\right)$.*

In [2], it was proven that, in several cases, linear index codes were optimal. They conjectured that for any side information graph G, the shortest-length index code would always be linear and have length ${minrk}_{2}\left(G\right)$. The conjecture was refuted by Lubetzky and Stav in [18]. In any case, as shown by Peeters [19], calculating the minrank of an arbitrary graph is a difficult task. More specifically, Peeters showed that deciding whether a graph has minrank three is an NP-complete problem.

**Example** **3.***Consider the instance of the index encoding problem given in Example 2. Then, we find that the matrix M that fits the hypergraph H has the form:*$$M=\left[\begin{array}{cccc} 1 & 0 & * & *\\ 0 & * & 1 & *\\ {}* & 0 & 0 & 1\\ 0 & * & 0 & 1\\ {}* & 1 & * & 0 \end{array}\right]$$*The lines are associated with the receivers $R_{1},\ldots ,R_{5}$ and the columns to the message indexes 1,2,3 and* 4. *The symbol “∗” can be replaced by an arbitrary element in the field $F_{q}$.*

For an example, consider the field $F_{2}$. A matrix that fits the hypergraph H has rank at least 3. Thus, we select
$$M=\left[\begin{array}{cccc} 1 & 0 & 0 & 0\\ 0 & 1 & 1 & 0\\ 0 & 0 & 0 & 1\\ 0 & 0 & 0 & 1\\ 0 & 1 & 1 & 0 \end{array}\right]$$
which achieves the minimum rank 3. Now, we consider three linearly independent lines of *M*, and suppose that
$$G=\left[\begin{array}{cccc} 1 & 0 & 0 & 0\\ 0 & 1 & 1 & 0\\ 0 & 0 & 0 & 1 \end{array}\right]\Rightarrow Gx=\left[\begin{array}{cccc} 1 & 0 & 0 & 0\\ 0 & 1 & 1 & 0\\ 0 & 0 & 0 & 1 \end{array}\right]\left[\begin{array}{c} x_{1}\\ x_{2}\\ x_{3}\\ x_{4} \end{array}\right]=\left[\begin{array}{c} x_{1}\\ x_{2}⊕x_{3}\\ x_{4} \end{array}\right]$$
The decoding process goes as follows. Since $R_{2}$ and $R_{5}$ already know $\left\{x_{2},x_{4}\right\}$ and $\left\{x_{1},x_{3}\right\}$, respectively, they obtain $x_{3}$ and $x_{2}$, respectively, from the first packet. Receiver $R_{1}$ obtains $x_{1}$ and both $R_{3}$ and $R_{4}$ obtain $x_{4}$.

**Remark** **2.**
*We have made available at [20] an algorithm (m-files) in Matlab, which is designed to solve small examples in this work, since, as we mentioned above, there is no polynomial-time algorithm for an arbitrary graph.*


### 2.4. Pliable Index Coding

The pliable index coding problem (PICOD), introduced by Brahma and Fragouli in [6], is a variant of the index coding problem. In PICOD, users do not have predetermined messages to decode, as in the case of classic index coding; instead, each user is satisfied to decode any message that is not present in its side information set. Figure 4 illustrates this system model.

The problem is formalized as follows: a transmitter with *n* messages $\left\{x_{i}:i\in \left[n\right]\right\}$, $x_{i}\in X$ is connected to *m* receivers $R_{1},\ldots ,R_{m}$, through a noiseless channel. Each receiver $R_{j}$ knows $x_{S_{j}}=\left\{x_{i}:i\in S_{j}\right\}$ as side information. We denote by $I_{j}≜\left[n\right]\setminus S_{j}$ the index set of the unavailable messages in $x_{S_{j}}$. Then, $x_{I_{j}}=\left\{x_{i}:i\in I_{j}\right\}$ denotes the set of requests of $R_{j}$. Each receiver $R_{j}$ is satisfied if it can successfully recover any message that is not present in its side information set, i.e., any message $x_{d}\in x_{I_{j}}$.

We can represent an instance of a pliable index coding problem using an undirected bipartite graph, one side representing the message indexes and the other side representing the receivers. We connect $R_{j}$ to the indices belonging to $I_{j}$, as in Figure 5.

**Remark** **3.**
*By having this freedom to choose the desired message for each user, PICOD can satisfy all users with a significant reduction in the number of transmissions compared to the index encoding problem with the same set of messages and the same sets of user side information.*


**Example** **4.**
*We will consider the case described in Example 2 as a pliable index coding problem. Now, we have the bipartite graph in Figure 5 describing the problem. Note, for example, that client 1 demands any of the messages indexed in $I_{1}=\left\{1,2\right\}$ and knows the indexed messages in $S_{1}=\left\{3,4\right\}$; client 3 will be satisfied to receive any of the messages $x_{2},x_{3}$ or $x_{4}$, since it only knows $x_{1}$.*


#### Pliable Index Coding via
Colorings of Hypergraphs

In [21], a graph coloring approach was presented for pliable index coding. The authors have shown the existence of a coding scheme that has length $O(\log^{2}\Gamma )$, where Γ refers to a hypergraph parameter that captures the maximum number of intersections between edges of the PICOD hypergraph.

An instance of the pliable index encoding problem is described by (m,n,I)−PICOD, onde $I=\{I_{j};j\in \left[m\right]\}$, and can be conveniently represented by a hypergraph.

**Definition** **9.**
*The hypergraph $H(V,E_{H})$, with V=[n] vertices and $E_{H}=\left\{e_{j}=\left(I_{j}\right);j\in \left[m\right]\right\}$ hyperedges, completely describes the (m,n,I)−PICOD. The hyperedge $e_{j}=\left(I_{j}\right)$ represents the set of requests for $R_{j}$ (i.e., $E_{H}=I$).*


The problem illustrated in Example 5 can be represented by a hypergraph, as can be seen in Figure 6.

Let $H=(V,E_{H})$ be a hypergraph and C:V→[L] be a coloring of V, where *L* is a positive integer. We say that *C* is a conflict-free coloring for the hyperdges, if each $E_{H}$ of H has at least one vertex with unique color. The smallest number of colors required for such a coloring is called the conflict-free chromatic number of H, denoted by $\chi_{CF}\left(H\right)$. This parameter was first introduced by Even et al. [22].

**Remark** **4.**
*In [21], pliable index coding was given a graph coloring approach. The authors have shown the existence of a coding scheme that has length $O(\log^{2}\Gamma )$, where Γ refers to a hypergraph parameter that captures the maximum number of “incidences” of other hyperedges in any given hyperedge. This result improves the best known achievable results, in some parameter regimes.*


**Definition** **10.**
*A pliable index code (PIC) consists of a collection of an encoding function on the server that encodes the n messages to an ℓ−length codeword,*

$$\phi :X^{n}⟶X^{\ell },$$

*and decoding functions $\psi_{j}:X^{\ell }\times X^{|S_{j}|}⟶X,$ satisfying $\psi_{j}\left(\phi \left(x\right),x_{S_{j}}\right)=x_{d},$ for some $d\in I_{j}$.*

*The quantity ℓ− is called the length of the pliable index code. We are interested in designing pliable index codes that have small ℓ−.*


We will assume that $X=F^{k}$ for some finite field F and integer k≥1. If k>1, we refer to this code as a k−*vector* PIC, while the k=1 case is also called a *scalar* PIC. We will concentrate on the linear PICs. In this case, the coding function ϕ is represented by a matrix ℓ×mk (denoted by G) such that $\phi \left(x_{i}:i\in \left[n\right]\right)=Gx^{T}$, where $x=(x_{11},\ldots ,x_{1k},\ldots ,x_{m1},\ldots ,x_{mk}).$ The smallest *ℓ* for which there is a linear k−*vector* PIC for an instance of the pliable index coding problem given by the hypergraph H will be denoted by $\ell_{k}^{*}\left(H\right)$.

**Definition** **11.**
*Let C:V→[L] be a conflict-free coloring of the hypergraph H that represents a PICOD. The indicator matrix associated with this coloring $G_{C}$, L×n, is given by*

$$G_{c}\left(c,i\right)=\left\{\begin{array}{cc} 1, & if\;the\;vertex\;ireceived\;the\;color\;c;\\ 0, & otherwise. \end{array}\right.$$



**Teorema** **1**([21]). *The indicator matrix $G_{c}$ generates the pliable index code for the problem described by the hypergraph H.*

**Example** **5.**
*Consider the same PICOD represented in Figure 6. The coloring shown in Figure 7 is a conflict-free coloring with two colors. Then, the matrix $G_{c}=\left[\begin{array}{cccc} 1 & 0 & 0 & 1\\ 0 & 1 & 1 & 0 \end{array}\right].$*


From the messages $x_{1}⊕x_{4}$ and $x_{2}⊕x_{3}$, all receivers can successfully recover at least one message from their request set.

Using the same parameters as in Example 2, we see that the length of the index code for this instance is ℓ=3, while, for the PICOD case, $\ell_{k}^{*}\left(H\right)=2$.

### 2.5. Index Coding via MDS Codes

The index coding model defined in Section 2.3, via graph theory, is only one of many approaches used to describe and solve an index encoding problem. One of the most interesting index coding schemes using codes has the maximum distance separable (MDS) property, which consists of transmitting κ(G)=n−mindeg(G), the parity symbols of a systematic MDS code with parameters (n+κ,n), where mindeg(G) represents the minimum amount of side information available at each receiver, i.e., for a general index encoding problem with side information graph G,
$$\mathrm{mindeg}\left(G\right)≜\min_{i\in [n]}\left|\left\{j;\left(i,j\right)\in E\left(G\right)\right\}\right|=\min_{i\in [n]}\left|S_{i}\right|.$$
Then, every receiver has *n* code symbols (including its side information) and, by the MDS property, it can successfully recover its desired message.

**Proposition** **1**([1]). *Consider an index coding problem with n messages and n receivers represented by the side information graph G. Let $S_{i}$, the side information set of the receiver $R_{i},i\in \left[n\right]$, and then*



$${\mathrm{minrk}}_{q}\left(G\right)\leq \kappa \left(G\right)=n-\mathrm{mindeg}\left(G\right)=n-\min_{i\in [n]}\left|S_{i}\right|.$$



**Corollary** **1.**
*If G is a complete graph, then mindeg(G)=n−1 and the transmission of the parity symbol $x_{1}+\cdots +x_{n}$ of an (n+1,n) MDS code over $F_{2}$ achieves minrk${}_{q}\left(G\right)=1$.*


## 3. Results

The tree construction method proposed in [12] can be interpreted as a network coding problem with multiple sources and multiple users. In the proposed model, both encoding and decoding are performed by polynomial operations, without the need for side information; however, if they exist, they may allow multiple interpretations at the receivers, based on the side information available at each receiver. Figure 8 illustrates this system model.

Given the connection between network and index coding problems, established in [4], we can also interpret the coding with nested cyclic codes, at the stage where the packets are XOR-ed together, as a case of MDS index coding according to Corollary 1, in the particular case where each receiver is unaware of the message it is requesting, which may be a rare occurrence. However, it is possible to take advantage of the method’s distinguishing feature—the possibility of multiple interpretations at the receivers—and, by imposing some extra conditions, design an index code model that has greater flexibility over the side information sets.

In the next subsections, we present some results and algorithm implementations, and in Section 4, we present in detail the proposed index encoding with multiple interpretations.

### 3.1. Index Coding from Reed–Solomon Codes

We establish a connection between index coding and error-correcting codes based on the tree construction method of nested cyclic codes proposed in [12]. We implement a few algorithms to perform tree construction using the Matlab language, which allows us to work over finite bodies in a practical and efficient way and helps to solve some implementation problems encountered later in [12]. We prove that for cyclic codes, there will not always be an increase in error correction capability between the levels of the tree, as suggested in [12]. This is why we have initially limited this study to Reed–Solomon codes, because they are MDS codes, which guarantees an increase in Hamming distance at each level, meaning that, under certain conditions, the knowledge of side information will be interpreted as an increase in the decoder’s ability to correct errors.

#### A Tree Construction with Nested Cyclic Codes

A nested code is characterized by a global code where each element is given by a sum of codewords, each belonging to a different subcode. That is,

$$c=i_{1}G_{1}⊕i_{2}G_{2}⊕\cdots ⊕i_{N}G_{N},$$
where ⊕ represents an XOR operation. For an information vector $i_{\ell },1\leq \ell \leq N$, the codeword $i_{\ell }G_{\ell }$ belongs to a subcode $C_{\ell }$ of code C and c∈C.

Nested cyclic codes, whose subcodes are generated by generator polynomials, were originally proposed by Heegard [23], and were originally called partitioned linear block codes. They can be defined as follows:

**Definition** **12.**
*Let $C=\{C\left(x\right)\in F_{q}\left[x\right];g\left(x\right)|C\left(x\right)\}$ be a t-error-correcting cyclic code having g(x) as the generator polynomial. Note that C=〈g(x)〉 is an ideal of the ring $R_{n}=F_{q}\left[x\right]/\left(x^{n}-1\right)$, but is also a vector subspace of $F_{q}^{n}$, such that*

$$C\left(x\right)=p_{1}\left(x\right)g_{1}\left(x\right)+p_{2}\left(x\right)g_{2}\left(x\right)+\cdots +p_{N}\left(x\right)g_{N}\left(x\right),$$

*where $C_{\ell }\left(x\right)=p_{\ell }\left(x\right)g_{\ell }\left(x\right)$, 1≤ℓ≤N is an encoded packet belonging to the $t_{\ell }$-error-correcting subcode*

$$C_{\ell }=\left\{C_{\ell }\left(x\right)\in F_{q}\left[x\right];g_{\ell }\left(x\right)|C_{\ell }\left(x\right)\right\},$$

*generated by $g_{\ell }\left(x\right)$ and satisfying the following conditions:*


*1*.
*$g_{\ell }\left(x\right)|g_{\ell +1}\left(x\right)$;*
*2*.
*$\deg\left[C_{\ell }\left(x\right)\right]<\deg\left[g_{\ell +1}\left(x\right)\right]$.*


The tree-based algebraic construction of nested cyclic codes, proposed in [12], aims to

1.Encode, independently, different data packets, providing protection against channel errors;2.Encode different data packets producing codewords that are added, resulting in the packet $C_{0}$;3.Correct the errors on $C_{0}$ and, finally, recover the data in the receiver by polynomial operations.

Consider a tree in which the root node is associated with the vector subspace of an encompassing error correcting code. Thus, the root node is defined as the code $C_{i0}$, such that
$$C_{i0}=\langle g_{i0}\left(x\right)\rangle =\left\{C_{i0}\left(x\right)\in F_{q}\left[x\right];g_{i0}\left(x\right)|C_{i0}\left(x\right)\right\}.$$

This subspace corresponds to a $t_{0}$-error-correcting cyclic code $C_{i0}\left(n,k_{i0}\right)$, generated by the polynomial $g_{i0}\left(x\right)$.

**Definition** **13.**
*A tree of nested cyclic codes is a finite tree such that*


*1*.
*Each inner node (including the root node) can be subdivided into another inner node and a terminal node;*
*2*.
*The jth th inner node is associated with a linear subspace $C_{ij}\subset F_{q}^{n}$ of dimension $k_{ij}$, and can be subdivided into the subspaces*

$$C_{ij}=C_{i(j+1)}+C_{t(j+1)}\mathrm{com}C_{i(j+1)}\cap C_{t(j+1)}=\left\{0\right\}e\;k_{ij}=k_{i(j+1)}+k_{t(j+1)}$$

*3*.
*The subspace $C_{ij}$, associated with the jth inner node, must be a cyclic linear block code generated by $g_{ij}\left(x\right)$;*
*4*.
*If $C_{ij}=\langle g_{ij}\left(x\right)\rangle$ e $C_{i(j+1)}=\langle g_{i(j+1)}\left(x\right)\rangle$, then $g_{ij}\left(x\right)|g_{i(j+1)}\left(x\right)$; furthermore, $g_{ij}\left(x\right)|x^{n}-1$ for any $g_{ij}\left(x\right)$;*
*5*.
*To conclude, the last inner node will have no ramifications.*


**Remark** **5.**
*Figure 9 illustrates the model described above.*


Let $p_{j}\left(x\right)$ be the data packet associated with the terminal node, for 1≤j≤T. The encoding is given by
$$C_{j}\left(x\right)=p_{j}\left(x\right)g_{i(j-1)}\left(x\right).$$
Then, the encoded packets are summed up and the resulting codeword is sent out by the transmitter
$$C_{0}\left(x\right)=C_{1}\left(x\right)+C_{2}\left(x\right)+\cdots +C_{T}\left(x\right).$$
After the error correction phase, the *j*th packet $p_{j}\left(x\right)$ is decoded by the operations:(1)$$p_{j}\left(x\right)=\left\{\begin{array}{cc} \left[C_{0}\left(x\right)\mathrm{mod}g_{ij}\left(x\right)\right]/g_{i(j-1)}\left(x\right) & \mathrm{if}1\leq j\leq T,\\ C_{0}\left(x\right)/g_{i(T-1)}\left(x\right) & \mathrm{if}j=T. \end{array}\right.$$

The information will be contained in the remainder of the division of $C_{0}\left(x\right)$ by $g_{ij}\left(x\right)$, since the modulo operation eliminates the influence of all messages related to polynomials of degree equal to or greater than the degree of $g_{ij}$. Thus, the quotient of the final division operation provides the desired information, since all other messages have degree less than the degree of the divisor polynomial. Therefore, in the case of the last package, only the division operation is required. We suggest consulting [12] for more details on the encoding process using the tree construction method.

### 3.2. Tree Construction: Algorithm and Considerations

We describe a few algorithms in Matlab and considerations for fitting to the model of tree construction, which can be found at [20], allowing us to perform the calculations on finite fields by making the appropriate transformations from integer representation to powers of α. Below, we exemplify the main idea of the algorithm.

**Example** **6.**
*For T=3 let $C_{i0}\left(7,5\right)$ be a Reed–Solomon code in GF(8) and $k_{t1}=k_{t2}=2$ the dimensions of subspaces $C_{t1},C_{t2}$, respectively. They are associated with the terminal nodes of the tree; the last node of the tree, which is an inner node without ramification, is associated with $C_{i2}$ of dimension $k_{i2}=1$.*


The packets $p_{1}\left(x\right)=x+\alpha^{2},p_{2}\left(x\right)=\alpha^{3}x+\alpha$, both associated with the the terminal nodes, have length 2; $p_{3}\left(x\right)=\alpha^{5}$ has length equal to 1 and is associated with the last node. Let α be the primitive element of GF(8), and the generator polynomials are
$\deg\left(g_{i0}\left(x\right)\right)=n-k_{i0}=2$$\Rightarrow g_{i0}\left(x\right)=\prod_{j=1}^{2}\left(x-\alpha^{j}\right)=x^{2}+\alpha^{4}x+\alpha^{3};$$\deg\left(g_{i1}\left(x\right)\right)=n-k_{i1}=4$$\Rightarrow g_{i1}\left(x\right)=\prod_{j=1}^{4}\left(x-\alpha^{j}\right)=x^{4}+\alpha^{3}x^{3}+x^{2}+\alpha x+\alpha^{3};$$\deg\left(g_{i2}\left(x\right)\right)=n-k_{i2}=6$$\Rightarrow g_{i2}\left(x\right)=\prod_{j=1}^{6}\left(x-\alpha^{j}\right)=x^{6}+x^{5}+x^{4}+x^{3}+x^{2}+x+1.$

Then, the encoded packets are



$\begin{array}{ccc} C_{1}\left(x\right) & = & p_{1}\left(x\right)g_{i0}\left(x\right)\\ & = & x^{3}+\alpha x^{2}+\alpha^{4}x+\alpha^{5}; \end{array}$



$\begin{array}{ccc} C_{2}\left(x\right) & = & p_{2}\left(x\right)g_{i1}\left(x\right)\\ & = & \alpha^{3}x^{5}+\alpha^{5}x^{4}+\alpha^{6}x^{3}+\alpha^{2}x^{2}+x+\alpha^{4}; \end{array}$



$\begin{array}{ccc} C_{3}\left(x\right) & = & p_{3}\left(x\right)g_{i2}\left(x\right)\\ & = & \alpha^{5}x^{6}+\alpha^{5}x^{5}+\alpha^{5}x^{4}+\alpha^{5}x^{3}+\alpha^{5}x^{2}+\alpha^{5}x+\alpha^{5}. \end{array}$



The transmitted codeword $C_{0}\left(x\right)$ is given by



$\begin{array}{ccc} C_{0}\left(x\right) & = & C_{1}\left(x\right)+C_{2}\left(x\right)+C_{3}\left(x\right)\\ & = & \alpha^{5}x^{6}+\alpha^{2}x^{5}+0x^{4}+\alpha^{3}x^{3}+1x^{2}+0x+\alpha^{4}. \end{array}$



**Remark** **6.**
*Each terminal node is a shortened version of the code associated with the inner node from which the terminal node emanates. It is implicit that the codewords of shortened codes are prefixed with zeros to achieve length n and, therefore, that these codes are not cyclic.*


#### 3.2.1. Decoding—Error Correction

Considering tree construction based on Reed–Solomon codes and assuming that the receiver has side information available, when will there be an increase in error correction capability?

**Proposition** **2.**
*Due to the nesting structure, the variable error correctability characteristic can only be observed if there is a sequential removal of the packets associated with the nodes from the root to the top of the tree.*


**Proof.** Supposing that $C_{\ell }\left(x\right),1\leq \ell \leq T$ is the first coded packet known at the receiver, then
$$\begin{array}{cc} C_{0}\left(x\right) & =p_{1}\left(x\right)g_{i0}\left(x\right)+\cdots +p_{\left(\ell -1\right)}\left(x\right)g_{i(\ell -2)}\left(x\right)+p_{\left(\ell +1\right)}\left(x\right)g_{i\ell }\left(x\right)+\cdots +p_{T}\left(x\right)g_{i(T-1)}\left(x\right)\\ & =\left[p_{1}\left(x\right)+\cdots +p_{\left(\ell -1\right)}\left(x\right)q_{\left(\ell -1\right)}\left(x\right)+p_{\left(\ell +1\right)}\left(x\right)q_{\left(\ell +1\right)}\left(x\right)+\cdots +p_{T}\left(x\right)q_{T}\left(x\right)\right]g_{i0}\left(x\right), \end{array}$$
therefore, $C_{0}\left(x\right)\in C_{i0}\left(n,k_{io}\right)$, whose error correction capability is $t_{0}$. Note that even though the receiver knows about other packages $C_{j}\left(x\right),\ell <j\leq T$, the result does not change. On the other hand, if all packages $C_{j}\left(x\right);1\leq j<\ell$ are known to the receiver, we can write
$$\begin{array}{cc} C_{0}\left(x\right) & =p_{\left(\ell +1\right)}\left(x\right)g_{i\ell }\left(x\right)+\cdots +p_{T}\left(x\right)g_{i(T-1)}\left(x\right)\\ & =\left[p_{\left(\ell +1\right)}\left(x\right){\bar{q}}_{\left(\ell +1\right)}\left(x\right)+\cdots +p_{T}\left(x\right){\bar{q}}_{T}\left(x\right)\right]g_{i\ell }\left(x\right), \end{array}$$
thus, $C_{0}\left(x\right)\in C_{i\ell }\left(n,k_{i\ell }\right)$, whose error correction capability is $t_{\ell }⩾t_{0}$, and equality occurs only when
$$d_{min}\left(C_{\ell }\right)-d_{min}\left(C_{0}\right)<2.$$□

**Example** **7.**
*Consider the same tree as in Example 6.*



*If all packages are unknown ⇒ the decoding is performed by $C_{i0}\left(7,5\right)$∴$t_{0}=1$;*

*If $C_{1}\left(x\right)$ is known ⇒ the decoding is performed by $C_{i1}\left(7,3\right)$∴$t_{1}=2$;*

*If $C_{2}\left(x\right)$ is known $C_{1}\left(x\right)\Rightarrow$ the decoding is performed by $C_{i2}\left(7,1\right)$∴$t_{2}=3$.*



*However, if $C_{2}\left(x\right)$ is known but $C_{1}\left(x\right)$ is not, then the resulting codeword still belongs to $C_{0}\left(x\right)\in C_{i0}\left(7,5\right)$, and there is no improvement in error correction capability, since*

$$\begin{array}{cc} C_{0}\left(x\right) & =p_{1}\left(x\right)g_{i0}\left(x\right)+p_{3}\left(x\right)g_{i0}\left(x\right)\bar{q}\left(x\right)\\ & =\left[p_{1}\left(x\right)+p_{3}\left(x\right)\bar{q}\left(x\right)\right]g_{i0}\left(x\right). \end{array}$$



Another advantage of Reed–Solomon codes is that they are easily decoded using an algebraic method known as syndrome decoding.

#### Syndrome Decoding

Syndrome decoding is an algebraic method based on the Berlekamp–Massey algorithm, which became a prototype for the decoding of many other linear codes.

If the coded package $C_{1}\left(x\right)$ is known and an error e(x) occurs, then the message received will be
$$\begin{array}{cc} r(x) & =C_{0}\left(x\right)+C_{1}\left(x\right)+e\left(x\right) \end{array}$$
Suppose that the error is given by $e\left(x\right)=0x^{6}+0x^{5}+\alpha^{2}x^{4}+\alpha^{5}x^{3}+0x^{2}+0x+0$, and then
$$\begin{array}{cc} r(x) & =\alpha^{5}x^{6}+\alpha^{2}x^{5}+\alpha^{2}x^{4}+\alpha^{6}x^{3}+\alpha^{3}x^{2}+\alpha^{4}x+1. \end{array}$$

**Remark** **7.**
*Notice that we need to find the error locations and their values, which is the main difference with binary codes, since, for binary codes, it is enough to determine the error locations.*


The decoding process can be divided into three stages.

1.Syndrome calculation

The syndrome calculation stage consists of checking the roots of the generating polynomial as inputs of r(x). If the result is null, the sequence belongs to the set of codewords and, therefore, there are no errors. Any nonzero value indicates the presence of an error.

If the encoded packet $C_{1}\left(x\right)$ is known, then the error correction algorithm is executed by $C_{i1}$, which is a RS(7,3) code, generated by $g_{i1}\left(x\right)=\left(x-\alpha \right)\left(x-\alpha^{2}\right)\left(x-\alpha^{3}\right)\left(x-\alpha^{4}\right)$. Then,
$$\begin{array}{cc} r(x) & =m\left(x\right)g_{i1}\left(x\right)+e\left(x\right), \end{array}$$
Therefore, evaluating the roots of g(x) at r(x), the result will only be null when there are no errors in the transmission.
$$∴S_{i}=r\left(x\right){|}_{x=\alpha^{i}}=e\left(\alpha^{i}\right),\forall \;i=1,\ldots ,n-k=2t.$$



$$\begin{array}{ccc} S_{1}=r\left(\alpha \right)=\alpha^{5}; & & S_{3}=r(\alpha^{3})=0;\\ S_{2}=r\left(\alpha^{2}\right)=\alpha^{6}; & & S_{4}=r\left(\alpha^{4}\right)=\alpha^{6}. \end{array}$$



2.Error Localization

Let μ be the number of errors 0≤μ≤t, which occur at locations $\ell_{1},\cdots ,\ell_{\mu }$, and the error polynomial can be written as
$$e\left(x\right)=e_{\ell_{1}}x^{\ell_{1}}+\cdots +e_{\ell_{\mu }}x^{\ell_{\mu }}.$$
To correct r(x), we must find the values and locations of the errors, which are denoted, respectively, by $e_{\ell_{1}},\ldots ,e_{\ell_{\mu }}$ and $x^{\ell_{1}},\ldots ,x^{\ell_{\mu }}$. Substituting $\alpha^{j},1\leq j\leq 2t$, into the error polynomial e(x), we obtain
$$\left\{\begin{array}{c} \begin{array}{ccccc} S_{1} & = & e(\alpha ) & = & e_{\ell_{1}}\alpha^{\ell_{1}}+e_{\ell_{2}}\alpha^{\ell_{2}}+\cdots +e_{\ell_{\mu }}\alpha^{\ell_{\mu }}\\ S_{2} & = & e(\alpha^{2}) & = & e_{\ell_{1}}{\left(\alpha^{\ell_{1}}\right)}^{2}+e_{\ell_{2}}{\left(\alpha^{\ell_{2}}\right)}^{2}+\cdots +e_{\ell_{\mu }}{\left(\alpha^{\ell_{\mu }}\right)}^{2}\\ & \vdots & \\ S_{2t} & = & e(\alpha^{2t}) & = & e_{\ell_{1}}{\left(\alpha^{\ell_{1}}\right)}^{2t}+e_{\ell_{2}}{\left(\alpha^{\ell_{2}}\right)}^{2t}+\cdots +e_{\ell_{\mu }}{\left(\alpha^{\ell_{\mu }}\right)}^{2t} \end{array} \end{array}\right..$$
Obtain $X_{i}=\alpha^{\ell_{i}}$ and $Y_{i}=e_{\ell_{i}}$ for 1≤i≤μ, where $X_{i}$ and $Y_{i}$ will represent, respectively, the locations and values of the errors. Note that we will have 2t equations and 2t unknowns, *t* being error values and *t* being locations.
$$\left\{\begin{array}{c} \begin{array}{ccc} S_{1} & = & Y_{1}X_{1}+Y_{2}X_{2}+\cdots +Y_{\mu }X_{\mu }\\ S_{2} & = & Y_{1}{X_{1}}^{2}+Y_{2}{X_{2}}^{2}+\cdots +Y_{\mu }X_{\mu }^{2}\\ & \vdots & \\ S_{2t} & = & Y_{1}{X_{1}}^{2t}+Y_{2}{X_{2}}^{2t}+\cdots +Y_{\mu }X_{\mu }^{2t} \end{array} \end{array}\right..$$
It can be shown that this nonlinear system has a unique solution if 0≤μ≤t [15]. The techniques that solve this system of equations include defining the error locator polynomial (ELP) σ(z) [24].

**Definition** **14.**
*Define the error locator polynomial sigma(z), as*




$$\begin{array}{ccc} \sigma (z) & = & \left(1-X_{1}z\right)\left(1-X_{2}z\right)\cdots \left(1-X_{\mu }z\right)\\ & = & \sigma_{\mu }z^{\mu }+\cdots +\sigma_{2}z^{2}+\sigma_{1}z+1. \end{array}$$

*The inverse of the square root of σ(z), $1/X_{1},\ldots ,1/X_{\mu }$, indicates the locations of errors.*


To find error locations $X_{i},1\leq i\leq \mu$, note that $\sigma_{1},\sigma_{2},\ldots ,\sigma_{\mu }$ and calculate the zeros of σ(z); to find them, we use a syndrome matrix, as we see below:$$\begin{array}{cc} \left[\begin{array}{cccc} S_{1} & S_{2} & \cdots & S_{\mu }\\ S_{2} & S_{3} & \cdots & S_{\mu +1}\\ \vdots & \vdots & \ddots & \vdots \\ S_{\mu } & S_{\mu +1} & \cdots & S_{2\mu -1} \end{array}\right]\left[\begin{array}{c} \sigma_{\mu }\\ \sigma_{\mu -1}\\ \vdots \\ \sigma_{1} \end{array}\right] & =\left[\begin{array}{c} -S_{\mu +1}\\ -S_{\mu +2}\\ \vdots \\ -S_{2\mu } \end{array}\right] \end{array}$$
Returning to Code RS(7,3), where the error correction capability is t=2, we must find $\sigma_{1}$ e $\sigma_{2}$:$$\begin{array}{cc} \left[\begin{array}{cc} S_{1} & S_{2}\\ S_{2} & S_{3} \end{array}\right]\left[\begin{array}{c} \sigma_{2}\\ \sigma_{1} \end{array}\right] & =\left[\begin{array}{c} S_{3}\\ S_{4} \end{array}\right]\\ \left[\begin{array}{cc} \alpha^{5} & \alpha^{6}\\ \alpha^{5} & 0 \end{array}\right]\left[\begin{array}{c} \sigma_{2}\\ \sigma_{1} \end{array}\right] & =\left[\begin{array}{c} 0\\ \alpha^{6} \end{array}\right]\Rightarrow \sigma_{2}=1e\sigma_{1}=\alpha^{6} \end{array}$$
Thus, $\sigma \left(z\right)=z^{2}+\alpha^{6}z+1$ with roots $\alpha^{3}$ and $\alpha^{4}$, so there is an error at the locations $\alpha^{-3}=\alpha^{4}$ and $\alpha^{-4}=\alpha^{3}$. Then,
$$e\left(x\right)=e_{3}x^{3}+e_{4}x^{4}.$$

3.Determining the error values

Calculating $e\left(x\right)=e_{3}x^{3}+e_{4}x^{4}$ at the points α and $\alpha^{2}$, we can use the syndromes already obtained, $S_{1}$ and $S_{2}$, to determine the values of the errors, solving the following system:$$\left\{\begin{array}{c} \begin{array}{ccccc} S_{1} & = & e(\alpha ) & = & e_{3}\alpha^{3}+e_{4}\alpha^{4}\\ S_{2} & = & e(\alpha^{2}) & = & e_{3}\alpha^{6}+e_{4}\alpha^{8} \end{array} \end{array}\right.$$
Therefore, the error polynomial is given by $e\left(x\right)=\alpha^{5}x^{3}+\alpha^{2}x^{4}.$ Now, correcting the received word r(x), we have
$$\begin{array}{cc} C_{0}\left(x\right) & =r\left(x\right)+c_{1}\left(x\right)+e\left(x\right)\\ & =\alpha^{5}x^{6}+\alpha^{2}x^{5}+0x^{4}+\alpha^{3}x^{3}+1x^{2}+0x+\alpha^{4}. \end{array}$$

#### 3.2.2. Decoding—Data Recovery

**Example** **8.**
*For the cases in the previous examples, where T=3, the original data can be recovered as follows:*

$$\begin{array}{cc} •\;p_{1}\left(x\right) & =\frac{C_{0}\left(x\right)\mathrm{mod}g_{i1}\left(x\right)}{g_{i0}\left(x\right)}=\frac{p_{1}g_{i0}\mathrm{mod}g_{i1}}{g_{i0}\left(x\right)}=\frac{p_{1}\left(x\right)g_{i0}\left(x\right)}{g_{i0}\left(x\right)};\\ •\;p_{2}\left(x\right) & =\frac{C_{0}\left(x\right)\mathrm{mod}g_{i2}\left(x\right)}{g_{i1}\left(x\right)}=\frac{p_{1}g_{i0}\mathrm{mod}g_{i2}+p_{2}g_{i1}\mathrm{mod}g_{i2}}{g_{i1}\left(x\right)};\\ •\;p_{3}\left(x\right) & =\frac{C_{0}\left(x\right)}{g_{i2}\left(x\right)}=\frac{p_{1}\left(x\right)g_{i0}\left(x\right)}{g_{i2}\left(x\right)}+\frac{p_{2}\left(x\right)g_{i1}\left(x\right)}{g_{i2}\left(x\right)}+\frac{p_{3}\left(x\right)g_{i2}\left(x\right)}{g_{i2}\left(x\right)}; \end{array}$$



In summary, the module operation removes the branches above the node of interest and the division operation removes the branches below. Therefore, no side information is needed at the receiver in order to recover the data packets.

##### Will There Always Be an Increase in Error Correction Capability?

We analyze two cases of tree construction of nested cyclic codes, with the same parameters at each level. In one of them, we observe no increase in the error correction capability from the second to last internal node of the tree. This is due to the variety of possibilities of generating polynomials for a cyclic code of parameters (n,k). As a result, we demonstrate in Proposition 3 that, for Reed–Solomon codes, this feature of increasing capacity will be guaranteed provided that: $k_{ij}-k_{i(j+1)}\geq 2,\forall j=0,\ldots ,T-1.$

**Example** **9.**
*Let $C_{i0}\left(15,10\right)$ be a cyclic code in GF(2) and $k_{t1}=4,k_{t2}=2$ be the dimensions of the subspaces $C_{t1},C_{t2}$, respectively. The last node is associated with $C_{i2}$ with dimension $k_{i2}=4$. The construction is depicted in Figure 10.*


We consider the factorization:$$x^{15}-1=\left(1+x\right)\left(1+x^{3}+x^{4}\right)\left(1+x+x^{2}+x^{3}+x^{4}\right)\left(1+x+x^{2}\right)\left(1+x+x^{4}\right)$$

Case 1.
•$g_{i0}\left(x\right)=\left(1+x\right)\left(1+x^{3}+x^{4}\right)\;$⇒ $t_{0}=1;$•$g_{i1}\left(x\right)=g_{i0}\left(x\right)\left(1+x+x^{2}+x^{3}+x^{4}\right)\;$⇒ $t_{1}=2;$•$g_{i2}\left(x\right)=g_{i1}\left(x\right)\left(1+x+x^{2}\right)\;$⇒ $t_{2}=3.$

Case 2.
•$g_{i0}\left(x\right)=\left(1+x\right)\left(1+x+x^{4}\right)\;$⇒$t_{0}=1;$•$g_{i1}\left(x\right)=g_{i0}\left(x\right)\left(1+x^{3}+x^{4}\right)\;$⇒$t_{1}=2;$•$g_{i2}\left(x\right)=g_{i1}\left(x\right)\left(1+x+x^{2}\right)\;$⇒$t_{2}=2.$

**Remark** **8.**
*We have provided an m-file algorithm at [20], which can be run through Matlab and performs the operations described in Examples 8 and 9.*


**Proposition** **3.**
*Given a (n,k) Reed–Solomon code, which has minimum distance d=n−k+1, one can guarantee an increase in error correction capability at each level of the tree provided that $k_{ij}-k_{i(j+1)}\geq 2,\forall j=0,\ldots ,T-1$.*


**Proof.** We must prove that $t_{i(j+1)}\geq t_{ij}+1,\forall j=0,\ldots ,T-1$. For simplicity but without loss of generality, set j=0. If $k_{i0}-k_{i1}\geq 2$, then we can write:

$$\begin{array}{cc} \left(-d_{i0}+n+1\right)+d_{i1}-n-1 & \geq 2\\ d_{i1}-1 & \geq d_{i0}-1+2\\ \left[\frac{d_{i1}-1}{2}\right] & \geq \left[\frac{d_{i0}-1}{2}\right]+1\\ t_{i1} & \geq t_{i0}+1. \end{array}$$
This completes the proof. □

The verification that, for cyclic codes, there will not always be an increase in the error correction capacity between the levels of the tree, as considered in [12], leads us to search for answers on how to properly choose the generating polynomials for a code of parameters (n,k) and its subcodes, in order to guarantee subcodes with larger Hamming distance, with the purpose of observing an increase in the error correction capacity between the levels of the tree. An approach to constructing chains of some linear block codes while keeping the minimum distances (of the generated subcodes) as large as possible is presented in [25] and may be the solution to this problem.

### 3.3. An Example with a BCH Code

According to Luo and Vinck [25], to construct a chain of BCH subcodes with the characteristic of maintaining the minimum distance as large as possible, the task becomes more difficult because their subcodes may not be BCH and cyclic codes, and therefore the minimum distance of these subcodes might not be found easily. However, for primitive BCH codes, the minimum distance coincides with the weight of the generator polynomial, which makes it feasible to use it for the construction of the nested subcode chain that we seek. For non-primitive BCH codes, this statement is not always valid. For an extensive description of the minimum distance for BCH codes, we recommend consulting [26].

In Table 1, we present the parameters for binary primitive BCH codes of length $n=2^{m}-1$; it will guide the tree construction.

**Example** **10.**
*Consider the root node associated with the BCH code $C_{i0}\left(15,11\right)$. Suppose that we want to encode the packets $p_{1}\left(x\right)=x^{3}+x^{2}+x$, $p_{2}\left(x\right)=x$ and $p_{3}\left(x\right)=x^{4}+x^{2}+1$ associated with nodes whose dimensions are $k_{t1}=4$, $k_{t2}=2$ and $k_{i2}=5$, respectively. The polynomials $g_{i0}\left(x\right)=x^{4}+x+1$, $g_{i1}\left(x\right)=x^{8}+x^{7}+x^{6}+x^{4}+1)$ and $g_{i2}\left(x\right)=x^{10}+x^{8}+x^{5}+x^{4}+x^{2}+x+1$ generate the codes associated with the internal nodes, $C_{i0}\left(15,11\right)$, $C_{i1}\left(15,7\right)$ e $C_{i2}\left(15,5\right)$, respectively, as shown in Figure 11.*


Encoding the packets, we have:$\begin{array}{ccc} C_{1}\left(x\right) & = & p_{1}\left(x\right)g_{i0}\left(x\right)\\ & = & x^{7}+x^{6}+x^{5}+x^{4}+x; \end{array}$$\begin{array}{ccc} C_{2}\left(x\right) & = & p_{2}\left(x\right)g_{i1}\left(x\right)\\ & = & x^{9}+x^{8}+x^{7}+x^{5}+x; \end{array}$$\begin{array}{ccc} C_{3}\left(x\right) & = & p_{3}\left(x\right)g_{i2}\left(x\right)\\ & = & \alpha^{5}x^{6}+\alpha^{5}x^{5}+\alpha^{5}x^{4}+\alpha^{5}x^{3}+\alpha^{5}x^{2}+\alpha^{5}x+\alpha^{5}. \end{array}$

The transmitted codeword $C_{0}\left(x\right)$ is given by:$$\begin{array}{ccc} C_{0}\left(x\right) & = & C_{1}\left(x\right)+C_{2}\left(x\right)+C_{3}\left(x\right)\\ & = & x^{14}+x^{8}+x^{7}+x^{6}+x^{3}+x+1 \end{array}$$

Alternatively, it is possible to represent the codeword in vector form:$$C_{0}\left(x\right)=\left(1,0,0,0,0,0,1,1,1,0,0,1,0,1,1\right)$$

After the error correction process, which will be performed on the sum of the coded packets, taking into account the side information available at each receiver, data recovery will occur as follows:$$\begin{array}{cc} •\;p_{1}\left(x\right) & =\frac{\left(x^{14}+x^{8}+x^{7}+x^{6}+x^{3}+x+1\right)\mathrm{mod}\left(x^{8}+x^{7}+x^{6}+x^{4}+1\right)}{\left(x^{4}+x+1\right)};\\ •\;p_{2}\left(x\right) & =\frac{\left(x^{14}+x^{8}+x^{7}+x^{6}+x^{3}+x+1\right)\mathrm{mod}\left(x^{10}+x^{8}+x^{5}+x^{4}+x^{2}+x+1\right)}{\left(x^{8}+x^{7}+x^{6}+x^{4}+1\right)};\\ •\;p_{3}\left(x\right) & =\frac{\left(x^{14}+x^{8}+x^{7}+x^{6}+x^{3}+x+1\right)}{\left(x^{10}+x^{8}+x^{5}+x^{4}+x^{2}+x+1\right)}. \end{array}$$

**Remark** **9.**
*We have made available at [20] an m-file Matlab algorithm that performs the tree construction operations and the data recovery for the BCH code.*


## 4. Index Coding with Multiple Interpretations

In the problem of index coding with multiple interpretations, we assume that receivers demand all the messages from the source and that the sender is unaware of the subset of messages already known in the receivers—performing an encoding so that any side information may be used by the decoder, in order to increase its error correction capability. Otherwise, if a receiver has no side information, the decoder considers the received word to belong to the highest rate code associated with the root node of the tree.

The proposed encoding process is shown in Figure 12 and can be performed in four main steps:1.Encoding of the different data packets with nested cyclic codes, which consists of subdividing the vector space of a linear block code into vector subspaces, using each of them for encoding a different user;2.Implementation of index coding at the relay node; the basic idea is that the different data packets, encoded by polynomial multiplications with linearly independent generators, are added and then forwarded to the receivers;3.Multiple interpretations at the receivers that occur at the error correction stage, where each receiver can decode the received message at different rates depending on the known side information;4.The data recovery stage, i.e., the process of decoding $C_{1}\left(x\right),\ldots ,C_{T}\left(x\right)$ through polynomial operations (Equation 1), as described in Section 3.1.

The notion of multiple interpretations was introduced in [13], indicating that the error correction capability in decoding gradually improves as the amount of side information available at the receiver increases. However, as we prove in Proposition 2, because of the nested structure of the tree, this characteristic of variable error correction capability can only be observed if there is a sequential removal of packets associated with the nodes, i.e., the side information should occur sequentially from the root to the top of the tree. However, in practice, this is not always the case. Thus, if we want to ensure that any information can be used efficiently in the decoder, it will be necessary to assume knowledge of the side information by the relay node or even the demand set, if we have a PICOD problem.

The following is a proposal for pliable index coding with multiple interpretations.

### Pliable Index Coding with Multiple Interpretations

As in the pliable index coding problem [6], we will assume that the transmitter knows the demand set of each receiver and that all receivers are satisfied by receiving any message contained in their demand set. For example, if we are searching on the internet for a red flower image and we already have some previously downloaded pictures on our computer, if we find any other image that we do not have yet, we will be satisfied.

The goal of the server is to find an encoding scheme that satisfies all receivers, using as few transmissions as possible and ensuring that all side information associated with nodes located below the node where the packet to be recovered is located may be interpreted as a gain in error correction capability, even when they do not appear in such a sequence.

The idea behind this proposal is to apply conflict-free coloring to the hypergraph that represents the demands of all receivers, and instead of sending the encoded word $C_{0}\left(x\right)=C_{1}\left(x\right)+C_{2}\left(x\right)+\cdots +C_{T}\left(x\right)$, we select the packets in a way that maximizes the possibility of a gain in error correction capability, since, as mentioned above, packages will only be removed if they occur sequentially.

**Example** **11.**
*Consider an instance of an pliable index coding with multiple interpretations in Figure 13a, where the encoded packets $C_{1}\left(x\right)=g_{i0}\left(x\right)p_{1}\left(x\right),C_{2}\left(x\right)=g_{i1}\left(x\right)p_{2}\left(x\right),C_{3}\left(x\right)=g_{i2}\left(x\right)p_{3}\left(x\right)$ and $C_{4}\left(x\right)=g_{i3}\left(x\right)p_{4}\left(x\right)$ will be sent to receivers $R_{1},R_{2}$ and $R_{3}$, which have demand sets $I_{1}=\left\{1,2\right\}$, $I_{2}=\left\{3\right\}$, $I_{1}=\left\{2,4\right\}$, respectively, as we see in Figure 13b.*


Figure 14 shows conflict-free coloring with two colors and $G_{c}=\left[\begin{array}{cccc} 1 & 0 & 1 & 1\\ 0 & 1 & 0 & 0 \end{array}\right]$, which represents the pliable index code.

Note that if we send only the message $C_{0}=C_{1}⊕C_{3}⊕C_{4}$, all receivers recover one and only one message from their request set, as we can see in Table 2.

Depending on the problem, this would be an ideal solution, since the transmitter may want each receiver to decode *only one message*, in which case we would have a PICOD(1); no client can receive more than one message from its request set. The case of PICOD(1) is dealt with in detail in [27], and the following example, which aptly illustrates its use, is provided.

Consider a media service provider whom we pay for movies. The provider has a set of movies and customers pay for a certain number of movies, e.g., one movie. Suppose that the service is being sold in such a way that customers will be happy to receive any movie that they have not watched yet. There is a restriction on the service provider’s side, since customers who have paid for only one movie should not receive more than one. Therefore, it can only supply one film for each client.

## 5. Conclusions

The verification that, for cyclic codes, there will not always be an increase in the error correction capacity between the levels of the tree leads us to search for ways to correctly choose the generating polynomials for a code and its subcodes, in order to guarantee subcodes with larger Hamming distance and an increase in error correction capability in consecutive levels of the tree. A method for the construction of chains of some linear block codes that maintains the minimum distances (of the generated subcodes) as large as possible is presented in Vinck and may be useful in addressing this issue.

Our work deals with the construction of index coding. We treat index coding as a network coding problem and we show how it is possible to construct pliable index codes with multiple interpretations by exploiting the conflict-free coloring of a hypergraph. Studying conflict-free coloring of a hypergraph in the context of the general index coding problem seems to be an interesting direction for future studies.

## Figures and Tables

**Figure 1 entropy-24-01149-f001:**
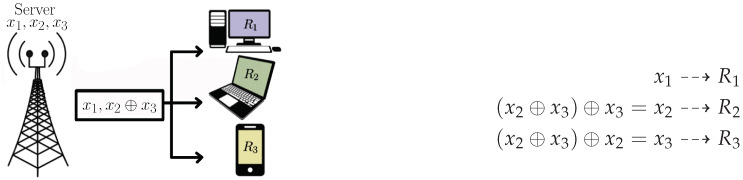
Index coding problem with three receivers.

**Figure 2 entropy-24-01149-f002:**
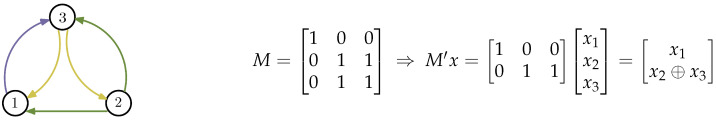
Graph and matrix related to the problem described in Figure 1.

**Figure 3 entropy-24-01149-f003:**
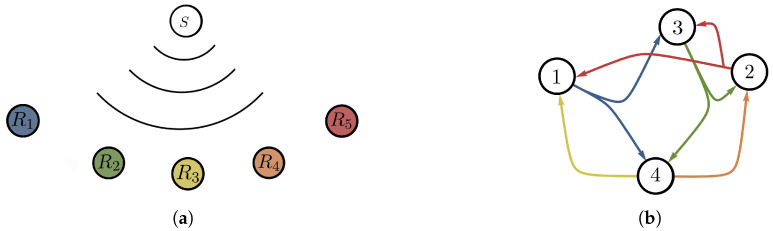
A single sender with multiple receivers having side information: (**a**) An instance of an index coding problem with m=5 (receivers) and n=4 (messages). (**b**) The hypergraph that describes this instance will have four vertices and five hyperedges: $e_{1}=\left(1,\left\{3,4\right\}\right)$, $e_{2}=\left(3,\left\{2,4\right\}\right)$, $e_{3}=\left(4,\left\{1\right\}\right)$, $e_{4}=\left(4,\left\{2\right\}\right)$ and $e_{5}=\left(2,\left\{1,3\right\}\right)$.

**Figure 4 entropy-24-01149-f004:**
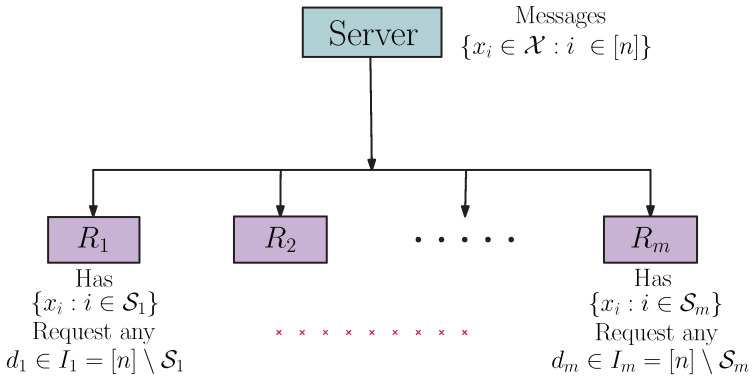
Pliable index coding scheme.

**Figure 5 entropy-24-01149-f005:**
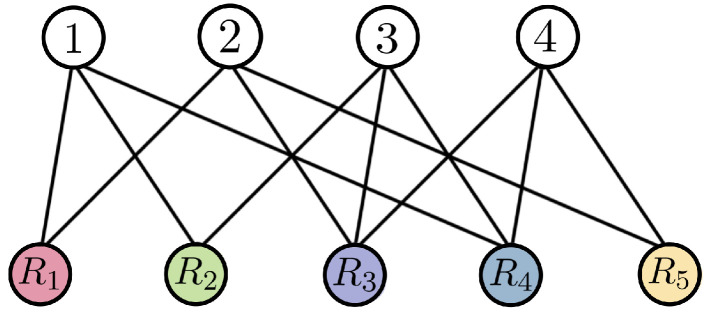
Bipartite graph for PICOD.

**Figure 6 entropy-24-01149-f006:**
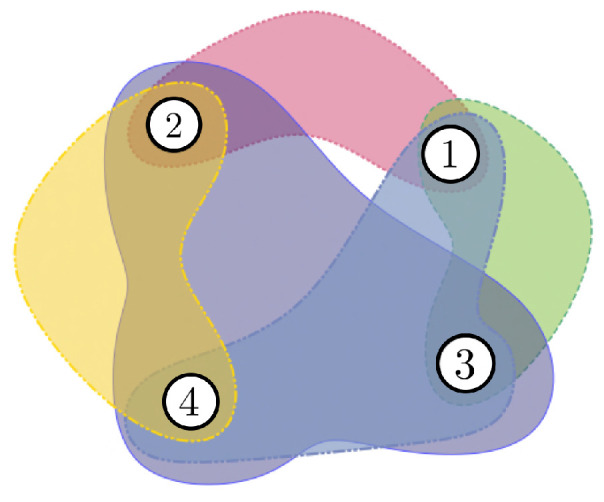
Hypergraph.

**Figure 7 entropy-24-01149-f007:**
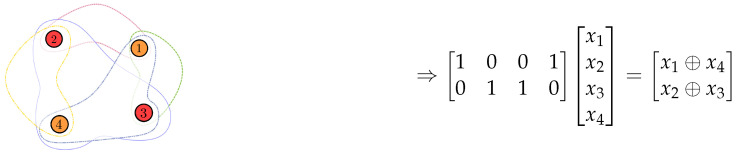
Conflict-free coloring with two colors.

**Figure 8 entropy-24-01149-f008:**
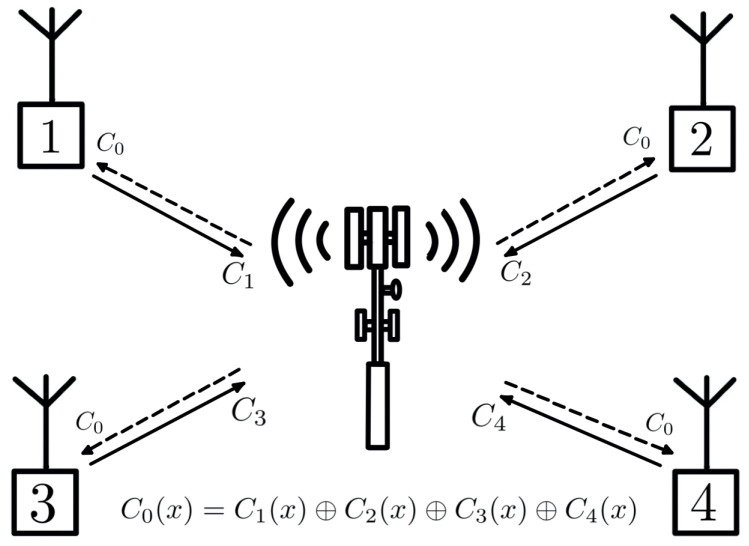
Coding model with nested cyclic codes.

**Figure 9 entropy-24-01149-f009:**
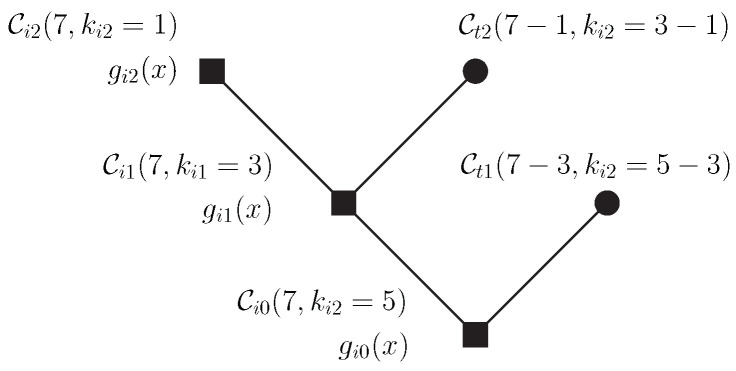
Tree construction. The sum of the dimensions associated with the last node and the terminal nodes is equal to the dimension of the root node.

**Figure 10 entropy-24-01149-f010:**
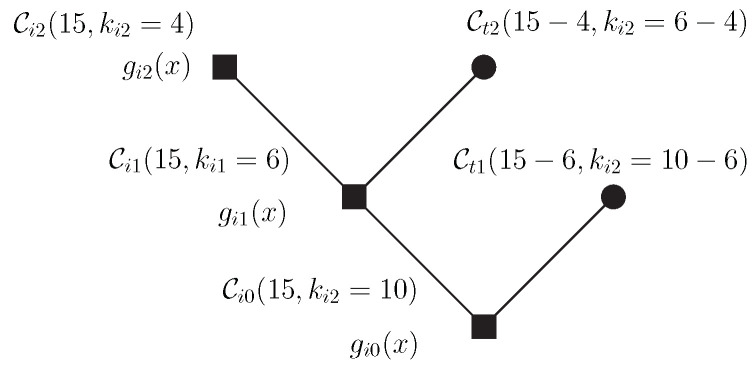
Tree construction.

**Figure 11 entropy-24-01149-f011:**
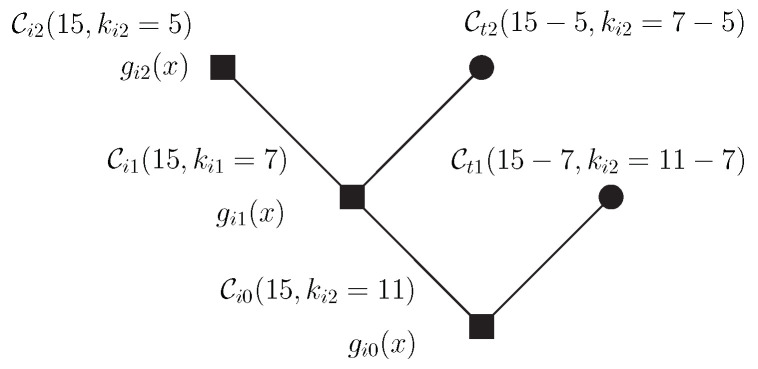
Tree construction of a BCH code tree.

**Figure 12 entropy-24-01149-f012:**
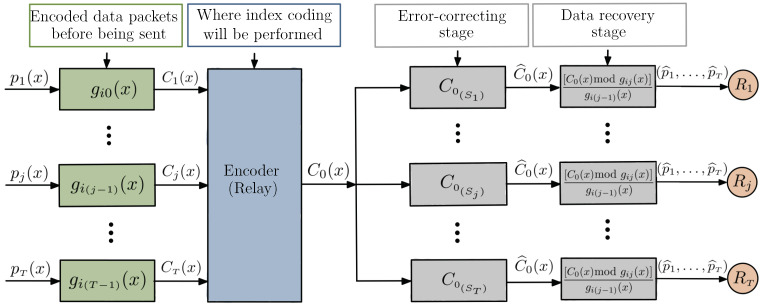
Scheme for index coding with multiple interpretations.

**Figure 13 entropy-24-01149-f013:**
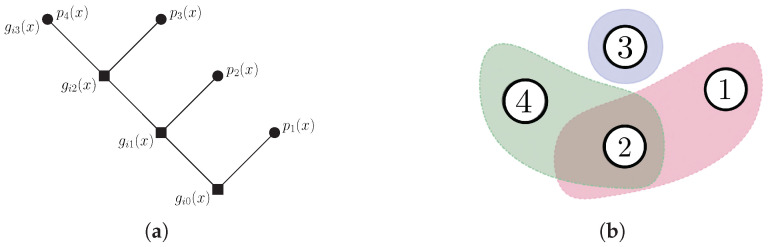
Pliable index coding with multiple interpretations: (**a**) The construction representation of an instance of a pliable index coding problem with m=3 (receivers) and n=4 (messages). (**b**) Shows the hypergraph that describes this instance.

**Figure 14 entropy-24-01149-f014:**
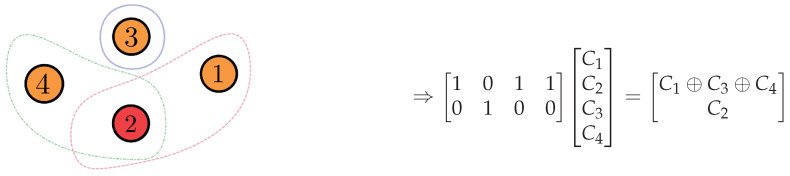
Conflict-free coloring with two colors.

**Table 1 entropy-24-01149-t001:** Parameters for values of m≤6.

		m=3		m=4				m=5						m=6										
*n*		7		15				31						63										
*k*		4		11	7	5		26	21	16	11	6		57	51	45	39	36	30	24	18	16	10	7
*t*		1		1	2	3		1	2	3	5	7		1	2	3	4	5	6	7	10	11	13	15

Note that there will always be an increase in error correction capability for a fixed *n* and varying *k*.

**Table 2 entropy-24-01149-t002:** Receivers with their side information sets.

Receivers	Side Information Sets	Decodes from Transmission
$R_{1}$	$S_{1}=\left\{3,4\right\}$	$C_{1}⊕C_{3}⊕C_{4}⊕C_{3}⊕C_{4}=C_{1}$
$R_{2}$	$S_{2}=\left\{1,2,4\right\}$	$C_{1}⊕C_{3}⊕C_{4}⊕C_{1}⊕C_{4}=C_{3}$
$R_{3}$	$S_{3}=\left\{1,3\right\}$	$C_{1}⊕C_{3}⊕C_{4}⊕C_{1}⊕C_{3}=C_{4}$

Each receiver *R_j_* is satisfied if it can successfully recover any new message that is not present in its side information set, i.e., any message *x_d_* ∈ $x_{I_{j}}$, where $I_{j}≜\left[n\right]\setminus S_{j}$.

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
