# Peer review of "Index Coding with Multiple Interpretations"

_entropy, 2022, doi:10.3390/e24081149_

Round 1

Reviewer 1 Report

Please see attached report.

Reviewer 2 Report

This work establishes  a connection between index coding and error-correcting codes with multiple interpretations from the tree construction of nested cyclic codes. It shows some examples to show its effectiveness. I have comments as follows:

1. The work combine index coding with network coding that was also stuided before. Please compare this perspective with the existing studies.

2. The work seems not be generalized and the result from information perspective should be explained.

3. How this scheme can be performed when different noise or erasure is considered.

Round 2

Reviewer 1 Report

Thanks for addressing the issues raised in my earlier report.

Reviewer 2 Report

 The authors has addressed all my concernes in the reply. The paper can be accepted.